# Analyzing the Influence of COVID-19 on the E-Commerce Customer's Retail Experience in the Supermarket Industry: Insights from Brazil

**Beatriz Moschetta Cunha, Carolina Kato Lettieri, Giulia Wiltenburg Cadena and Veridiana Rotondaro Pereira *** 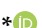

Department of Production Engineering, Mackenzie Presbyterian University, São Paulo 01302-907, Brazil; bia.mcunha@hotmail.com (B.M.C.); carolkato12@hotmail.com (C.K.L.); giuliawcadena@gmail.com (G.W.C.)
* Correspondence: veridiana.pereira@mackenzie.br

**Abstract:** *Background*: E-commerce's convenience, speed, and ability to overcome geographical barriers have made it increasingly popular across industries. This study analyzed the impact of the COVID-19 pandemic on customer experience and satisfaction in supermarket retail e-commerce in the Brazilian market. The objective was to understand how the pandemic influenced satisfaction in this sector. *Methods*: A survey research method was employed, utilizing the Critical Incident Technique to identify key quality dimensions important to customers. A total of 133 valid responses underwent exploratory factor analysis. *Results*: Data analysis identified three dimensions significantly affecting overall customer satisfaction. "Presentation of Products in the Application (app)" had the highest correlation, while "Delivery Logistics" had the lowest. However, when considering the COVID-19 factor, "Delivery Logistics" showed the highest correlation. This indicates that the dimension most affected by the pandemic has the least impact on overall satisfaction in supermarket e-commerce. *Conclusions*: The study emphasizes the significance of understanding customer satisfaction in supermarket e-commerce, not just during the COVID-19 pandemic but also its post-pandemic consequences. Retailers should prioritize improving product presentation in apps, as it greatly influences overall satisfaction. Addressing the pandemic's impact on delivery logistics is also crucial to ensure competitiveness in the e-commerce market.

**Keywords:** e-commerce market; customer satisfaction; COVID-19 pandemic



## 1. Introduction

In a globalized world, the internet has become the main technology for communication and inclusion [1]. This has brought new business models, such as electronic commerce, or e-commerce. In e-commerce, all stages of the sales process, from the search for the product by the customer to delivery, can be done entirely online, eliminating distances and borders. According to Ahi et al. [2], e-commerce is an example of how business can take advantage of digital technologies. According to Nanda et al. [3], given the COVID-19 pandemic, the retail industry has undergone rapid adaptation, with digital technology-driven platforms playing a crucial role. This trend is expected to persist as both consumers and retailers continue to adjust to the new realities of the post-pandemic era.

E-commerce refers to the processing of transactions electronically conducted over internet, such as the buying and selling of products and services over computer networks like the internet [4,5]. It involves a set of activities that directly influence the strategic planning of an organization, creating a network of interaction with customers. With the bargaining power acquired due to the expansion of e-commerce, consumers have become more demanding and discerning regarding the services provided. As a result, understanding their behavior has become a relevant topic for online companies [6]. According to Grewal et al. [7], providing a satisfying customer experience is very important for a company's survival. The quality of services is directly responsible for generating satisfaction levels

that lead to customer loyalty and consequently higher profitability for organizations [8]. Thus, the company must satisfy its consumers to achieve a competitive advantage in the online market [9].

Doherty and Ellis-Chadwick [10] state that while retailing is still primarily focused on selling in physical stores, the large and ongoing migration to e-commerce has been accelerated by the new coronavirus pandemic [11–13]. What was expected to take years occurred in a few months because the social distancing imposed by the COVID-19 pandemic made consumers seek new ways of buying [3,13,14].

According to a survey released in 2020 by the Brazilian Electronic Commerce Association (ABCOMM), in partnership with the Buy & Trust Movement, in the months of January through August 2020, Brazilian e-commerce grew 56.8% compared to the same months in 2019. The same survey indicated that the number of transactions made in e-commerce grew 65.7%, from 63.4 billion to 105.6 billion in the first six months of 2020 [15]. Furthermore, a survey conducted in 2020 by Sebrae, in partnership with the Getúlio Vargas Foundation (FGV), concluded that about 15% of the micro and small businesses interviewed entered online sales because of the pandemic. About 50% of them were already selling before the pandemic and continued throughout this period, totaling approximately 66% of the small businesses studied [16].

The 42nd edition of Webshoppers, the most comprehensive e-commerce report in Brazil, compiled biannually, indicated that in the first half of 2020, 7.3 million consumers entered e-commerce market. This number is close to the same number of new Brazilians who started shopping online throughout the entire year of 2019. The survey also indicated that 95% of people intend to continue shopping online in 2021 [17]. According to data from the MCC-ENET index, which is a reference for metrics and indicators of online consumption in Brazil, there was a 55.74% increase in revenues and a 53.83% increase in e-commerce purchases in the month of December 2020 compared to the same period in 2019 [18].

Supermarket e-commerce, called e-grocery, was one of the fastest growing sectors in online retail in recent months because of the pandemic. A survey by Visa Consulting & Analytics conducted in 2020 identified that the supermarket sector recorded a 373% increase compared to the period April to June 2019 [19].

Thus, based on the evidence of exponential growth in online purchases, there was a pressing social need for a relevant study that focused on retailers and e-commerce stores, particularly those in the supermarket retail sector, as they could greatly benefit from such research. The present study aimed to analyze the impact of the COVID-19 pandemic on the customer experience in online grocery retail e-commerce and its direct influence on customer satisfaction. Our objective was to identify the key factors that customers prioritized when making online purchases and their expectations regarding the quality of service provided. Additionally, this research held academic significance and practical value as it explored the social behavior of e-commerce customers within the pandemic landscape, where people were increasingly turning to online platforms due to prevailing lifestyle trends.

## 2. Theoretical Background

### 2.1. Consumer Behavior

According to Lovelock and Wirtz [20], service is an act or performance offered by one party to another, describing its intangibility and simultaneous consumption. As services are consumed as they are produced and are tied to a company's competitiveness, they must be delivered with quality and geared to consumer behavior [21,22].

Thus, consumer behavior is defined as the study of how individuals select, purchase, and discard goods and services to satisfy their needs and desires [23], influenced by cultural, social, and personal factors, which can also be divided into internal and external variables that affect the consumer [24].

External variables are defined as situational and environmental (weather variables and the time of year), social and cultural (set of values and customs), and reference groups (pro-

fessional groups, family, and friends) [25]. Internal variables are classified as gender, age, personality, self-image, and lifestyles and are fundamental to the perception of consumers' personal decisions [25].

The increased participation of service companies in the economy throughout the 1980s lead to the realization of the link between product and service quality and customer satisfaction and company profitability [26]. Consequently, organizations begin to strive for excellence in the quality of their services and products, considered fundamental to conquer and maintain the market [27]. Thus, the quality in services is based on the difference between customers' expectation of the service to be provided and their perception of how it is provided [28,29]. This makes it a priority to identify the various criteria that define the quality of a particular service, i.e., those that meet customers' needs and expectations [21].

### 2.2. E-Commerce

In the last decade, the Brazilian consumer market has embraced e-commerce as the main driving force, which encompasses different categories, among them electronic retail. Turban and King [30] understand electronic retailing as retail transactions between companies and final individual buyers through computer networks or the internet, where the latter can purchase and/or complete the transaction through an electronic system. Retailing is understood as any economic activity of selling a good or a service to the final consumer [31].

According to the executive director of the Buy & Trust Movement, the increase in e-commerce in Brazil can be attributed to a shift in the behavior of Brazilian consumers. They have become more actively involved in remote purchasing and have shown a significant preference for consuming products that fulfill their daily essential needs [15].

### 2.3. COVID-19 Implications

In December 2019, the world began to face a major health crisis with the new corona virus (SARS-CoV-2), called COVID-19, which emerged in China and spread rapidly. Its rapid and exponential growth led the World Health Organization (WHO) to declare a global pandemic on 11 March 2020. Brazil was the first country in Latin America to present a case of COVID-19, on 25 February 2020 [32].

To contain the advance of the contamination, many countries adopted measures that included encouraging hygiene and the use of masks, social distancing, closing establishments, prohibiting large events and gatherings, restricting travel, and making the population aware of the need to stay home [32].

Due to the pandemic, the lives of people and companies in all sectors have undergone many changes. These include restrictions on the movement of consumers, workers, and consumer goods, impacting logistics, routine activities, and interpersonal interactions [13,33]. From this, changes in habits and behavior patterns that were in motion greatly accelerated. The dynamics of digitizing business and intensifying the use of digital channels are examples [14,33]. E-commerce has become an ally for businesses, who now have another option to sell products and services. Some of the key points that had a significant impact include:

*Explosive increase in demand*—A survey conducted by EY Parthenon and published by VEJA INSIGHTS in 2020 found that with the retail shutdown, 62% of Brazilians visited fewer physical stores and 32% increased online shopping for food, which had been one of the segments with little e-commerce participation. The survey also found that, among the respondents, 74% of them intended to do more online shopping. Finally, order deliverability gained importance for 68% of consumers, as has product availability [33].

Finally, according to the Brazilian Association of Electronic Commerce (AbComm) [15], online supermarket purchases registered an increase of 180% since March 2020. According to data in a report by Ebit|Nielsen [17], the number of new consumers in supermarket e-commerce practically doubled after the beginning of the quarantine, even though this sector was considered essential and was not required to close its doors during the pandemic.

*Change in consumer preferences*—Customers had to quickly adapt to new forms of online purchasing, with many experiencing them for the first time. Furthermore, the safety and reliability of delivery services and the availability of online payment options became key factors in customers' choices [34,35].

*Logistical challenges*—The sudden increase in demand overwhelmed supply chains and delivery services. Logistics companies had to handle a much higher volume of orders, leading to delivery delays and operational difficulties [12]. Additionally, additional safety and hygiene measures were required at distribution centers and in delivery processes, impacting the efficiency and speed of operations [36].

*Need for rapid adoption of digital solutions*—A remarkable and pervasive trend that cuts across various domains is the rapid pace of technological change and digitalization. With the advent of the Internet and mobile technologies, we are observing profound transformations that extend to all sectors and aspects of human life [3]. Certainly, the advent of online retail platforms or applications (apps) has propelled the retail sector into a new era. Enabled by communication technologies, customers now have access to products at highly competitive prices, while retailers benefit from reduced operating costs.

Online platforms play a crucial role in enabling retailers to minimize operating costs and enhance overall business efficiency [3]. By leveraging these platforms, retailers can streamline their operations, optimize resource allocation, and achieve cost savings [37]. The online environment offers opportunities for automation, streamlined processes, and increased productivity, leading to improved efficiency and effectiveness in retail operations [38]. It is equally important to emphasize the usability aspects of the application (app) that significantly contribute to the user's seamless experience in finding products effortlessly [3].

## 3. Material and Methods

### 3.1. Data Collection

The present study aimed to analyze the effects of the COVID-19 pandemic on the customer experience of supermarket retail e-commerce, and its impact on customer satisfaction in the Brazilian market. A quantitative approach was adopted to allow objective analysis of the data collected about consumers' opinions related to this type of service. Quantitative research focuses on objectivity, using mathematics and statistics to describe the causes of a phenomenon, the relationship between variables, etc. [39].

For the nature of the relationship between the variables, the research has a descriptive method to describe the characteristics of a certain population or the establishment of relationships between variables. One of its most significant features is the use of standardized techniques for data collection [40].

Thus, we chose to use the survey research method, which can be described as obtaining data on the characteristics, actions, or opinions of a certain group of people, who are considered representative of a target population, using a research instrument such as a questionnaire [25]. The research was divided into two stages. The first stage focused on determining the research instrument, designing the pilot questionnaire, and refining it by applying it to a target audience, specified according to the research objective. The second stage focused on the implementation of the final questionnaire, data analysis, and interpretation of the observed results, with presentation of the outcome.

The questionnaire was made available for data collection in the city of São Paulo, Brazil, during the period of 9/22/2021 to 10/13/2021, to ensure a sufficient sample size and timeframe for data analysis. It was shared via various channels, including social media platforms such as WhatsApp, to reach a diverse range of participants. A total of 136 responses were collected, providing a substantial dataset for the research.

### 3.2. Critical Incident Technique and Formulation of the Questionnaire

In the literature, one of the relevant models proposed for measuring quality services is the Critical Incident Technique. Developed by Flanagan [35], this technique leverages

customer input to determine their own needs. By gaining knowledge about customer perceptions and reactions, companies can make more informed strategic decisions. A critical incident refers to a specific instance of service or product delivery that indicates a positive or negative performance by the supplier as perceived by the customer [22].

In this study, we adopted the Critical Incident Technique developed by Flanagan [35] as a method to develop our research instrument, primarily due to its advantages. This approach actively involves the customer as a valuable source of information, allowing for flexibility in its implementation. It enables the identification of significant aspects based on customer input, rather than relying solely on predetermined standards set by a specific method or organization [22]. These aspects, as defined by consumers, can be understood as characteristics of the supermarket e-commerce service that represent crucial factors or dimensions of quality. This process helps us determine which dimensions of quality are most important in describing the service based on customer opinions [22].

The ideal number of respondents for the collection of critical incidents should be between 10 and 20 people, because the deficient information from one respondent could be supplied by another respondent [22]. Fifteen people were interviewed, and each participant was asked to indicate three positive and three negative points that they consider when making grocery purchases via application (app). In total, 97 critical incidents were collected and subsequently grouped according to similarity into satisfaction items as 27 positive statements. Then three questions were added to the questionnaire to assess overall satisfaction [22].

Additional questions related to gender and age group were also asked to stratify the sample and permit analysis for each of the groups.

The survey instrument was structured with the 30 assertive questions. Initially, three people took a pretest to provide their opinion on the questions before starting data collection [41], verify if they understood the questions clearly and unambiguously, and thus improve the questionnaire before publishing it. The respondents found nothing to change, and the same questionnaire and the three collected responses were maintained.

The questionnaire was set up with a 7-point Likert scale. According to Joshi, et al. [42], a 7-point scale offers a wider range of choices when compared with a 5-point Likert scale, thereby increasing the likelihood of capturing the objective reality of individuals. It provides the interviewee with greater autonomy to select the "exact" option that aligns best with their preferences, instead of settling for a nearby or similar option [43–45]. The lowest value on the scale represents a negative response (Strongly Disagree with this Statement) and the highest value represents a positive response (Strongly Agree with this Statement). The variability of the scores resulting from the scale permits evaluation of consumer opinion about application-based grocery shopping in the Brazilian market.

### 3.3. Premises for Data Analysis

To analyze the data collected in the questionnaire, the exploratory factor analysis (EFA) technique was used, which groups correlated satisfaction items into factors, considered dimensions within the data about the quality of service of retail e-commerce in the supermarket sector. The construction of the factor analysis model was structured in six stages [46].

The first stage is to define the summary of characteristics as the objective of the factor analysis, by applying a correlation matrix of the variables. The summary of the data collected by the questionnaire allows the set of variables to be observed at various levels of generalization, until the number of variable groupings (factors) adequately represent the original set of variables.

The second stage focuses on planning the analysis, in which a minimum sample of 100 collected answers was defined, following the general rule of the minimum number for conducting factor analysis [46].

The third stage consists of checking if the variables are sufficiently correlated to be considered representative factors. The Bartlett's sphericity test was used, which is based on

the chi-square statistical distribution. For the factor analysis method to be appropriate, the null hypothesis that the correlation matrix is an identity matrix must be rejected. In other words, the significance value of the Bartlett's test should be less than 0.05 ($p < 0.05$). The Measure of Sampling Adequacy (MSA) was also used; this ranges from 0 to 1, reaching 1 when each variable is perfectly predicted without error by the other variables. If the measure is not high, that variable can be excluded from the analysis. The overall MSA and the MSA of each variable should be greater than 0.5 [46]. For this work, both tests reached satisfactory levels to proceed with the results analysis, with $p < 0.01$ and 0.880, respectively.

The fourth stage focuses on extracting factors by using principal component analysis, a method best suited for research that focuses on summarizing the original information to a minimum number of factors. The criteria for extracting the factors were the eigenvalue and the percentage of accumulated variance. For the eigenvalue, each variable contributes a value of 1 to the total eigenvalue. The widely used rule (Kaiser's criterion) states that only factors with eigenvalues greater than 1 should be extracted. The reasoning behind this criterion is that eigenvalues represent the amount of variation explained by a factor and that an eigenvalue of 1 represents a substantial amount of variation (FIELD, 2009). For the criterion on the accumulated percentage variance, the extraction of factors continues until a specific plateau is obtained. Hair, Black, Babin, Anderson and Tatham [46] suggest 60% as an acceptable level in the humanities and social sciences. Thus, the extracted factors had a minimum eigenvalue of 1, and the accumulated variance was 61.80%.

In the fifth stage, the oblimin method was used to perform the oblique rotation of the factors by rotating the axis of the factors to achieve a simple and significant factorial pattern with correlation between inherent dimensions. The minimum factor loading level to be considered significant is 0.40 for the sample size [47]. The specificity (Uniqueness) is the part of the data variance that cannot be explained by the factor, so that the higher the specificity, the lower the relevance of the variable in the factorial model. The specificity must be lower than 0.5 [46].

For data analysis, Jamovi software was used [48]. This open-source statistical tool offers a user-friendly interface, ensuring accessibility and ease of use for researchers. Jamovi provides a comprehensive suite of statistical tests and analysis features, enabling us to conduct the necessary analyses to fulfill our research objectives.

## 4. Results and Discussion

After distributing the questionnaire, 136 responses were collected. Cases where respondents provided the same answers for all statements were identified and removed. This could be due to respondent inattention or lack of interest, rendering these responses unreliable. In such circumstances, the quality of the response is compromised, and it is not advisable to consider them for analysis [49]. Thus, these responses were removed from the survey, resulting in 133 valid responses.

The analysis started with an evaluation of specificity, represented by the "Uniqueness" measure, to validate the unexplained variance. Oblimin rotation was applied to all factors to determine if any questions exhibited a specificity above 0.5. Among the questions, 4, 13, 14, 18, 21, 23, 26, 27, 28, and 29 demonstrated a Uniqueness score higher than 0.5. Questions 13, 18, 21, 23, 27, and 28 were subsequently excluded from the analysis, while questions 14 and 26 were retained despite having a Uniqueness score of 0.505 and 0.501, respectively, due to their proximity to the cutoff threshold. Questions 4 and 29, related to COVID-19, were also retained since the COVID factor is essential for the research objective and cannot be eliminated. The factors in the final rotation are presented in Table 1, while Table 2 presents the COVID-19 factor.

**Table 1.** Factors and Uniqueness.

| | Component | | | |
|---|---|---|---|---|
| | **1** | **2** | **3** | **Uniqueness** |
| Q24 | 0.792 | | | 0.386 |
| Q22 | 0.782 | | | 0.322 |
| Q15 | 0.772 | | | 0.323 |
| Q12 | 0.766 | | | 0.268 |
| Q11 | 0.616 | | | 0.382 |
| Q8 | 0.539 | 0.474 | | 0.382 |
| Q26 | 0.430 | | | 0.501 |
| Q6 | | 0.761 | | 0.328 |
| Q10 | | 0.754 | | 0.263 |
| Q9 | | 0.751 | | 0.350 |
| Q5 | | 0.652 | | 0.473 |
| Q3 | | 0.638 | | 0.372 |
| Q17 | | | 0.689 | 0.412 |
| Q2 | | | 0.646 | 0.362 |
| Q16 | | | 0.635 | 0.360 |
| Q14 | | | 0.535 | 0.505 |
| Q7 | | | 0.488 | 0.497 |

Note. 'Oblimin' rotation was used. Source: Compiled by the authors (2022).

**Table 2.** COVID-19 Factor and Uniqueness.

| | Component | |
|---|---|---|
| | **1** | **Uniqueness** |
| Q19 | 0.810 | 0.345 |
| Q29 | 0.644 | 0.585 |
| Q4 | 0.598 | 0.643 |

Note. 'Oblimin' rotation was used. Source: Compiled by the authors (2022).

The present study identified three factors, along with a COVID-19 factor. Additionally, an extra factor was included to measure overall satisfaction, which encompasses customers' general satisfaction with the service and their perception of how they were treated by the company [22]. In total, the analysis considered five factors: the three identified factors, the COVID-19 factor, and the overall satisfaction factor. Table 3 provides a comprehensive overview of these factors, including their corresponding questions and assigned names.

**Table 3.** Description of scale used in this study.

| Factor | Name of the Factor | Question |
|---|---|---|
| 1 | Overall satisfaction | Q20. The services provided by the e-commerce marketplace are satisfactory. |
| | | Q25. I am satisfied with the services provided by the e-commerce marketplace. |
| | | Q30. The quality of service provided by the e-commerce marketplace meets my expectations. |
| 2 | COVID-19 | Q4. The pandemic influenced my choice to shop with an application (app). |
| | | Q19. I feel that I expose myself less to COVID. |
| | | Q29. I plan to use shopping application (app) even with the end of the pandemic. |
| 3 | Delivery Logistics | Q3. Home delivery is a facilitator. |
| | | Q5. The products are packaged appropriately. |
| | | Q6. It was quick to order from the application (app). |
| | | Q9. The selected products were within their expiration date. |
| | | Q10. The purchase performance time is adequate. |

**Table 3.** *Cont.*

| Factor | Name of the Factor | Question |
|---|---|---|
| 4 | Product Presentation on the Application (app) | Q22. It is easy to find the products in the application (app). |
| | | Q26. The chosen products were delivered correctly. |
| | | Q12. The application (app) provides items in an organized manner. |
| | | Q15. It is easy to find information about the products. |
| | | Q24. The application (app) has variety of product brands. |
| | | Q11. It is easy to interact with the application (app). |
| 5 | Delivery and Price | Q14. The prices in the application (app) are the same as in the physical markets. |
| | | Q2. The delivery was done within the promised time. |
| | | Q7. The freight is a reasonable value. |
| | | Q16. The selected products are in good condition. |
| | | Q17. The delivery is done quickly. |

Source: Compiled by the authors and translated from the original which was in Portuguese (2022).

Taking into consideration that the scale was derived from the critical incident technique [22], which focuses on identifying aspects that are important to customers rather than predefined standards, the factors in this study were named based on their general aspects.

Factor 3 was labeled as "Delivery Logistics" because it encompasses all the necessary steps and considerations involved in delivering products to customers' doorsteps, including facilitation, packaging, ordering speed, product freshness, and timely delivery performance [13].

Factor 4 was named "Product Presentation on the Application (app)" as it encompasses the aspects related to the design, organization, and usability aspects of the application (app) that contribute to the user's ability to find products easily [3], interact seamlessly, access relevant information, receive accurate deliveries, and explore a variety of product options.

Factor 5 "Delivery and Price" is related to several key aspects of the customers' perceptions and evaluations regarding the pricing of products [50], the timeliness and quality of deliveries, and the overall value they receive in terms of product condition and delivery efficiency [13].

Next, Cronbach's alpha analysis was performed to verify the reliability and measure the internal consistency of each factor, as Cronbach's alpha is the average of the correlations between the items that are part of the factors (STREINER, 2003). The measure ranges from 0 to 1, with values of 0.6 and 0.7 being the minimum for acceptance (HAIR Jr. et al., 2009). All factors obtained an alpha greater than 0.6, except for the factor "COVID-19", which will be kept, since it is the factor with the COVID-19 questions which is fundamental to the objective of the study (Table 4).

**Table 4.** Cronbach's alpha per factor.

| Factor | Cronbach's Alpha |
|---|---|
| Overall Satisfaction | 0.862 |
| COVID-19 | 0.411 |
| Delivery Logistics | 0.844 |
| Product Presentation on the Application (app) | 0.872 |
| Delivery and Price | 0.742 |

Source: Compiled by the authors (2022).

### 4.1. Result of the Sample Stratification

The questionnaire included questions to analyze the characteristics of the sample related to gender and age group. For gender, 18.05% of the respondents were male and 81.95% female. For the age group, the percentage distribution of respondents was 45.9%

for 18 to 24 years, 13.5% for 25 to 34 years, 16.5% for 35 to 45 years, and 24.1% for 46 to 60 years.

To analyze each factor, an average factor score was created to evaluate the level of overall satisfaction with the dimension proposed by the factor. Descriptive statistics for overall factor analysis and the Shapiro–Wilk normality test were performed (Table 5), and then each factor per sample group was analyzed.

**Table 5.** Descriptive statistics and Shapiro–Wilk normality test.

|  | Overall Satisfaction | COVID-19 | Delivery Logistics | Product Presentation on the Application (App) | Delivery and Price |
|---|---|---|---|---|---|
| N | 133 | 133 | 133 | 133 | 133 |
| Mean | 5.36 | 5.75 | 5.86 | 5.23 | 5.10 |
| Median | 5.67 | 5.67 | 6.00 | 5.17 | 5.20 |
| Standard deviation | 1.22 | 1.01 | 1.04 | 1.14 | 1.05 |
| Minimum | 1.67 | 2.33 | 1.00 | 2.00 | 2.20 |
| Maximum | 7.00 | 7.00 | 7.00 | 7.00 | 7.00 |
| Shapiro–Wilk p | <0.001 | <0.001 | <0.001 | 0.004 | 0.022 |

Source: Compiled by the authors (2022).

Overall, the mean factorial scores performed well, with the obtained means ranging from 5.10 to 5.86 within the 7-point scale. The factor "Delivery Logistics" had the highest average (5.86), indicating high consumer satisfaction with this factor, while "Delivery and Price" had the lowest average (5.10). Although this value does not indicate dissatisfaction, it suggests that improvement could be made to increase customers' perception of this factor. For all factors, the Shapiro–Wilk normality test was non-significant ($p < 0.05$); therefore, nonparametric tests were used to compare the populations.

The correlation analysis of the factors with "Overall Satisfaction" was performed using Pearson's correlation (Table 6), which showed that the factor "Presentation of Products in the Application (app)" had the highest degree of correlation and the factor "Delivery Logistics" had the lowest degree of correlation. The correlation analysis of the "COVID-19" factor found the "Delivery Logistics" factor had the highest degree of correlation.

**Table 6.** Correlation with overall satisfaction and COVID-19.

|  | Overall Satisfaction | | COVID-19 | |
|---|---|---|---|---|
|  | Pearson's r | *p*-Value | Pearson's r | *p*-Value |
| Overall satisfaction | - | - |  |  |
| COVID-19 | 0.626 | <0.001 | - | - |
| Delivery Logistics | 0.504 | <0.001 | 0.553 | <0.001 |
| Product Presentation on the Application (app) | 0.770 | <0.001 | 0.465 | <0.001 |
| Delivery and Price | 0.557 | <0.001 | 0.445 | <0.001 |

Source: Compiled by the authors (2022).

The factor "Product Presentation on the Application (app)" showed the highest correlation with "Overall Satisfaction". Note that the factor "Delivery Logistics", which showed the lowest correlation with "Overall Satisfaction", obtained the highest degree of correlation with "COVID-19", which indicates that the factor most impacted by the COVID-19 pan-demic had the least impact on the satisfaction of supermarket retail e-commerce customers.

4.1.1. Analysis by Gender

The gender groups had similar results between them. The averages obtained for females range from 5.01 to 5.80, while for males, they range from 5.34 to 6.12. The factor "Delivery Logistics" presented a higher average for both, which makes it more relevant. On the other hand, the factor "Delivery and Price" had a lower average for females, and the factor "Presentation of Products in the Application (app)" had a lower average for

males. To analyze whether the populations have significant differences, the Mann–Whitney (U) test was performed to test for equality of the medians (Table 7). The *p*-value indicates significance only for the factor "Delivery and Price", which demonstrates that only this factor presents disagreement of opinion between the genders.

**Table 7.** Mann–Whitney test (U).

|  |  | *p* |
|---|---|---|
| Overall Satisfaction | Mann–Whitney U | 0.962 |
| COVID-19 | Mann–Whitney U | 0.550 |
| Delivery Logistics | Mann–Whitney U | 0.315 |
| Product Presentation on the Application (app) | Mann–Whitney U | 0.540 |
| Delivery and Price | Mann–Whitney U | 0.037 |

Source: Compiled by the authors (2022).

### 4.1.2. Analysis by Age Group

For the age groups, the factor "Delivery Logistics" had the highest score for all age groups, which indicates that it is the most relevant factor for customer satisfaction for all ages. The factor with the lowest average score was "Delivery and Price" for ages 18 to 24, 25 to 34, and 46 to 60, while "Presentation of Products in the Application (app)" had the lowest score for ages 35 to 45. To analyze if the populations have significantly different answers, the Kruskal–Wallis test was performed (Table 8). The *p*-value was significant for all factors; therefore, post hoc analysis was performed (Table 9).

**Table 8.** Kruskal–Wallis Test.

|  | $X^2$ | df | *p* |
|---|---|---|---|
| Overall Satisfaction | 19.30 | 3 | <0.001 |
| COVID-19 | 10.88 | 3 | 0.012 |
| Delivery Logistics | 15.20 | 3 | 0.002 |
| Product Presentation on the Application (app) | 13.15 | 3 | 0.004 |
| Delivery and Price | 9.02 | 3 | 0.029 |

Source: Compiled by the authors (2022).

**Table 9.** Post hoc analysis for factors with significance.

| Factor | Pairwise Comparisons | | *p* |
|---|---|---|---|
| Delivery Logistics | 18 to 24 years | 35 to 45 years | 0.010 |
|  | 18 to 24 years | 46 to 60 years | 0.022 |
| Product Presentation on the Application (app) | 18 to 24 years | 35 to 45 years | 0.002 |
| Delivery and Price | 18 to 24 years | 35 to 45 years | 0.021 |

Source: Compiled by the authors (2022).

Statistically, the differences between the age groups 18 to 24 years and 35 to 45 years are significant for all three factors influencing "Overall Satisfaction", which may indicate different types of needs and requirements to achieve satisfaction according to age.

## 5. Conclusions

In this study, we have analyzed the impact of the COVID-19 pandemic on the customer experience in online grocery retail e-commerce, specifically focusing on its direct influence on customer satisfaction. From this analysis, we have gathered valuable information regarding our key findings, which will now be discussed in detail.

At first, a research instrument was elaborated based on the Critical Incident Technique developed by Flanagan [35]. This approach actively engages customers as valuable sources of information and allows for the identification of significant aspects based on their input. The survey instrument was structured with 30 assertive questions.

The second point is that three factors were identified as key components of quality supermarket retail e-commerce service: "Delivery Logistics", "Product Presentation on the Application (app)" and "Delivery and Price". Two other factors were included: "Overall Satisfaction" serving as a parameter for customer perception and "COVID-19" acting as a control variable for the study.

Finally, data analysis revealed that the factor most affected by "COVID-19" was "Delivery Logistics". Conversely, when considering "Overall Satisfaction", the strongest correlation was found with "Product Presentation on the Application (app)". This observation underscores the interesting finding that the aspect most impacted by the pandemic had the least influence on customer satisfaction within the realm of supermarket retail e-commerce.

## 6. Theoretical and Managerial Implications

Our objective was to identify the key factors that customers prioritize when making online purchases and understand their expectations regarding the quality of service provided. This paper contributes to the e-commerce literature in several ways. First, the development of a survey instrument was structured based on the significant aspects identified by customers. This instrument can be utilized to measure customer satisfaction with online services. Secondly, the findings of the study unveil that overall customer satisfaction in the supermarket retail e-commerce sector is not only significant but also satisfactory. These insights can significantly contribute to future studies in the e-commerce literature, providing valuable information for literature reviews and advancing our understanding of customer satisfaction in this specific industry. Finally, this study specifically focused on e-grocery, a rapidly expanding sector in online retail, particularly in recent months [19]. The insights gained from this research can serve as a valuable contribution to future studies in the field of e-commerce literature reviews. By examining the dynamics and customer satisfaction within the supermarket e-commerce sector, this study offers insights that can be relevant and applicable to other industries.

In addition to theoretical implications, the study offers practical contributions by exploring the social behavior of e-commerce customers within the pandemic landscape, where people were increasingly turning to online platforms due to prevailing lifestyle trends. Based on the observation that the aspect most affected by the pandemic had the least influence on customer satisfaction in the realm of supermarket retail e-commerce, it becomes evident that retailers in this sector need to prioritize the customer experience. This entails focusing on enhancing both the visual and functional aspects of their app to meet customer expectations and ensure greater satisfaction. A user-friendly design and intuitive navigation of the application empower users to browse and locate products with ease. By prioritizing usability, the app enhances the user's ability to navigate through the interface and quickly locate desired products and ultimately enhances their overall satisfaction with the shopping experience.

With the beginning of the pandemic, demand rapidly shifted from physical commerce to online, causing retailers to adapt their operations to create or consolidate e-commerce. This movement was followed by significant challenges, such as the need for adaptation to an explosive increase in demand, changes in consumer preferences, and logistical challenges and the need for rapid adoption of digital solutions.

These challenges are evident in the study results, which indicated a greater influence of COVID-19 on the "Delivery Logistics" factor. This task involved adapting the physical operation to include delivering products to customers' doorsteps, packaging, ensuring product freshness, and maintaining timely delivery performance.

The major challenge now for retailers in the supermarket e-commerce sector is understanding how to readjust their new operations to a post-pandemic reality. Some factors that can facilitate this challenge are adjustments made to supply chains, which had to become more flexible in response to unexpected changes in demand and product availability. On the other hand, operational costs, such as the policy of offering free shipping, are likely to be reevaluated based on the level of services provided. Another important issue is

ordering speed and for retailers, operational efficiency is crucial, and time is one of the most expensive components of last-mile delivery.

Lastly, it is worth noting for managerial implications that there are significant differences observed between the age groups of 18 to 24 years and 35 to 45 years in relation to all three factors that influence "Overall Satisfaction". This suggests the presence of distinct needs and requirements for retailers to achieve satisfaction based on age.

## 7. Limitations and Future Research

This study has several limitations. Firstly, the research sampling was based on convenience rather than a random selection, making it impossible to generalize the results. Secondly, the study focused exclusively on the city of São Paulo, Brazil. This deliberate choice was made to facilitate an in-depth analysis of the e-commerce customer experience within a specific urban context. However, it is important to note that the findings may not directly apply to other cities or regions. Thus, future research should aim to include a broader geographical scope to enhance the external validity of the study's findings. Additionally, for future studies, it is recommended to explore the analysis of post-pandemic impacts on freight policies and the various components of last-mile delivery.

**Author Contributions:** Conceptualization, B.M.C., C.K.L., G.W.C. and V.R.P.; methodology, B.M.C., C.K.L., G.W.C. and V.R.P.; software, B.M.C., C.K.L. and G.W.C.; validation, B.M.C., C.K.L., G.W.C. and V.R.P.; formal analysis, B.M.C., C.K.L., G.W.C. and V.R.P.; investigation, B.M.C., C.K.L. and G.W.C.; resources, B.M.C., C.K.L. and G.W.C.; data curation, B.M.C., C.K.L., G.W.C. and V.R.P.; writing—original draft preparation, B.M.C., C.K.L., G.W.C. and V.R.P.; writing—review and editing, B.M.C., C.K.L., G.W.C. and V.R.P.; visualization, B.M.C., C.K.L., G.W.C. and V.R.P.; supervision, V.R.P.; project administration, B.M.C., C.K.L. and G.W.C.; funding acquisition, V.R.P. All authors have read and agreed to the published version of the manuscript.

**Funding:** This research was funded by Mackenzie Research and Innovation Fund (MACKPESQUISA), grant number: 221061.

**Data Availability Statement:** Data sharing is not applicable to this article.

**Conflicts of Interest:** The authors declare no conflict of interest.

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
