# Peer review of "Analyzing the Influence of COVID-19 on the E-Commerce Customer’s Retail Experience in the Supermarket Industry: Insights from Brazil"

_logistics_

Round 1
Reviewer 1 Report
This paper explores the effects of the COVID-19 pandemic on the customer experience of supermarket retail e-commerce and its impact on customer satisfaction. The topic is interesting and timely. Here are my comments to improve the paper:
1- in the abstract, you need to include the major aspects of the entire paper in a prescribed sequence that includes: 1) the overall purpose of the study and the research problem(s) you investigated; 2) the basic design of the study; 3) major findings or trends found as a result of your analysis; and, 4) a brief summary of your interpretations and conclusions. At the moment, some of these aspects are missing.
2- In the introduction, make clearer what knowledge gaps you identified and how your research addresses them. Also, make the research objectives/questions clearer. Answer the “so what?” question. Why investigating such matter is important? End the introduction with an outline of the paper; what comes next?
3- related to the previous comment, the novelty/originality should be clearly justified that the manuscript contains sufficient contributions to the new body of knowledge from the international perspective. What new things (new theories, new methods, or new policies) can the paper contribute to the existing international literature? This point must be reasonably justified by a Literature Review, clearly introduced in Introduction Section, and completely discussed in Discussion Section.
4- You have overlooked some of the most recent and relevant publications on the impact of covid-19 on retail, make sure to include them in your literature review or introduction:
https://doi.org/10.1177/00420980221143043
https://doi.org/10.3390/su141811463
https://doi.org/10.33050/atm.v5i1.1497
5- how did you validate your data?
6- What are the limitations of your study?
7- You need to separate your discussion and conclusion sections. Discussion unrolls the main results, explain their meanings. Put there the new questions and perspectives, describe the most interesting points for the entire field. Define the possible answers, write down why and how and what for, your suggestions. Conclusion is a summary of the discussion or the whole work. You need to further elaborate on your conclusions. What do your results mean for policy makers?
8- you need to refer back to the literature and previous studies in your result, discussion and conclusion sections.
9- The conclusion could do more to tease out the wider resonance of the paper for the journal's international readership. At the moment, it is very premature.
10- I would like to see more of a reflection in the conclusion section. What are the implications of your work for future research, industry, and decision makers? There is room for improvement in your conclusion section.
Moderate editing of English language required
Author Response
Comments and Suggestions for Authors
This paper explores the effects of the COVID-19 pandemic on the customer experience of supermarket retail e-commerce and its impact on customer satisfaction. The topic is interesting and timely. Here are my comments to improve the paper:
1- in the abstract, you need to include the major aspects of the entire paper in a prescribed sequence that includes: 1) the overall purpose of the study and the research problem(s) you investigated; 2) the basic design of the study; 3) major findings or trends found as a result of your analysis; and, 4) a brief summary of your interpretations and conclusions. At the moment, some of these aspects are missing.
We would like to express our gratitude to the reviewer for bringing this to our attention.
The abstract has been revised to align with the prescribed sequence and word limit of 200 words set by the journal.
Background: E-commerce's convenience, speed, and ability to overcome geographical barriers have made it increasingly popular across industries. This study analyzed the impact of the COVID-19 pandemic on customer experience and sat-isfaction in supermarket retail e-commerce. The objective was to understand how the pandemic influenced satisfaction in this sector.
Methods: A survey research method was employed, utilizing the Critical Incident Technique to identify key quality di-mensions important to customers. A total of 133 valid responses underwent exploratory factor analysis.
Results: Data analysis identified three dimensions significantly affecting overall customer satisfaction. "Presentation of Products in the Application" had the highest correlation, while "Delivery Logistics" had the lowest. However, when con-sidering the COVID-19 factor, "Delivery Logistics" showed the highest correlation. This indicates that the dimension most affected by the pandemic has the least impact on overall satisfaction in supermarket e-commerce.
Conclusions: The study emphasizes the significance of understanding customer satisfaction in supermarket e-commerce, not just during the COVID-19 pandemic but also its post-pandemic consequences. Retailers should prioritize improving product presentation in applications, as it greatly influences overall satisfaction. Addressing the pandemic's impact on delivery logistics is also crucial to ensure a seamless customer experience and maintain competitiveness in the e-commerce market.
2- In the introduction, make clearer what knowledge gaps you identified and how your research addresses them. Also, make the research objectives/questions clearer. Answer the “so what?” question. Why investigating such matter is important? End the introduction with an outline of the paper; what comes next?
Thank you for your valuable feedback. We have carefully considered your suggestions and revised address the mentioned points. In the introduction, we have provided a clearer explanation of the knowledge gaps identified and how our research fills those gaps. We have also refined the research objectives and questions to explicitly state the purpose of our investigation. Regarding the importance of this study, we have emphasized the significance of understanding the impact of the COVID-19 pandemic on customer experience in the context of online grocery retail.
3- related to the previous comment, the novelty/originality should be clearly justified that the manuscript contains sufficient contributions to the new body of knowledge from the international perspective. What new things (new theories, new methods, or new policies) can the paper contribute to the existing international literature? This point must be reasonably justified by a Literature Review, clearly introduced in Introduction Section, and completely discussed in Discussion Section.
We thank the reviewer for your positive assessment. We have made a significant effort to address all your comments.
4- You have overlooked some of the most recent and relevant publications on the impact of covid-19 on retail, make sure to include them in your literature review or introduction:
We appreciate the reviewer's comment regarding the inclusion of recent and relevant publications on the impact of COVID-19 on retail. We acknowledge the importance of staying up to date with the latest research in the field. In response to this feedback, we incorporated all suggestions and some other recent and pertinent publications on the subject into our literature review and introduction sections.
https://doi.org/10.1177/00420980221143043
White, J.T.; Hickie, J.; Orr, A.; Jackson, C.; Richardson, R. The experience economy in UK city centres: A multidimensional and interconnected response to the ‘death of the high street’? Urban Studies 0, 00420980221143043, doi:10.1177/00420980221143043.
https://doi.org/10.3390/su141811463
Lashgari, Y.S.; Shahab, S. The Impact of the COVID-19 Pandemic on Retail in City Centres. Sustainability 2022, 14, 11463.
https://doi.org/10.33050/atm.v5i1.1497
Sayyida, S.; Hartini, S.; Gunawan, S.; Husin, S.N. The Impact of the Covid-19 Pandemic on Retail Consumer Behavior. APTISI Transactions on Management (ATM) 2021, 5, 79-88.
New reference
Nanda, A.; Xu, Y.; Zhang, F. How would the COVID-19 pandemic reshape retail real estate and high streets through acceleration of E-commerce and digitalization? Journal of Urban Management 2021, 10, 110-124, doi:https://doi.org/10.1016/j.jum.2021.04.001.
Gulfraz, M.; Sufyan, M.; Mustak, M.; Salminen, J.; Srivastava, D. Understanding the impact of online customers’ shopping experience on online impulsive buying: A study on two leading E-commerce platforms. Journal of Retailing and Consumer Services 2022, 68, 103000, doi:10.1016/j.jretconser.2022.103000.
Sayyida, S.; Hartini, S.; Gunawan, S.; Husin, S.N. The Impact of the Covid-19 Pandemic on Retail Consumer Behavior. APTISI Transactions on Management (ATM) 2021, 5, 79-88, doi:10.33050/atm.v5i1.1497.
Matuszelański, K.; Kopczewska, K. Customer Churn in Retail E-Commerce Business: Spatial and Machine Learning Approach. J. Theor. Appl. Electron. Commer. Res. 2022, 17, 165-198.
5- how did you validate your data?
The data analysis in our study was validated based on the premises indicated in the "Premises for data analysis" section. These premises served as the foundation for ensuring the reliability and accuracy of our data. We carefully followed the specified criteria and guidelines during the data analysis process to maintain the integrity of the findings.
6- What are the limitations of your study?
7- You need to separate your discussion and conclusion sections. Discussion unrolls the main results, explain their meanings. Put there the new questions and perspectives, describe the most interesting points for the entire field. Define the possible answers, write down why and how and what for, your suggestions. Conclusion is a summary of the discussion or the whole work. You need to further elaborate on your conclusions. What do your results mean for policy makers?
8- you need to refer back to the literature and previous studies in your result, discussion and conclusion sections.
9- The conclusion could do more to tease out the wider resonance of the paper for the journal's international readership. At the moment, it is very premature.
10- I would like to see more of a reflection in the conclusion section. What are the implications of your work for future research, industry, and decision makers? There is room for improvement in your conclusion section.
We appreciate the suggestions 6 to 10. It has been taken into consideration as the Conclusion section has been revised to reflect the impacts for the retail industry resulting from the findings and explanation on the limitations was added into the study.
Comments on the Quality of English Language
Moderate editing of English language required
Thank you for your feedback. We appreciate your input regarding the need for moderate editing of the English language in our paper.

Reviewer 2 Report
Dear authors, this an interesting paper and topic, but I would recommend some suggestions to improve the quality of this manuscript:
1º. Authors do not follow the journal's rules in the entire paper. For instance, authors need to provide in the abstract section: objectives, methods, findings, and new contribution in this area of knowledge. In the structure of the abstract section need to add: Background; Methods; Results... Furthermore, from my point of view, authors should change the title because this is focused in Brazil, possibly it is a case of study.
2º. Keywords should include e-commerce and Retail experience words.
3º. I reviewed the entire reference and I saw that this manuscript requires more updated studies, authors are facing a topic which technologies, ubiquity, information, communication change every day through new devices and processes. Indeed, authors claims this in the first paragraph: "In a globalised world, the internet has become the main technology for communication and inclusion"
4º. Introduction section has to tackle the main gaps of this research, why authors have developed this study, the main objectives and research questions, and where this study was done. In Brazil, Mexico... readers need to know this information in the introduction part, and authors did no face this information here. Indeed, there are a lot information that should be included in the literature review.
5º Authors need to explain this acronym: MCC-ENET.
6º. Theoretical background must provide the main keywords to stage the narrative of the paper, objectives, and research questions. For example, in the first paragraphs of the Literature review, authors wrote mix different concepts like: SERVICE; CONSUMER BEHAVIOUR; EXTERNAL VARIABLES.. Authors need to add a sub-section to explain each concept that is related to your objectives and questions. It is really confuse for readers and researchers. I recommend authors include these authors:
Florido-Benítez, L. (2016). Airport Mobile Marketing as a Channel to Promote Cross-selling. Journal Airline and Airport Management, 6(2), 133-151. https://upcommons.upc.edu/bitstream/handle/2117/99719/59-425-1-PB.pdf
Gulfraz, M. B., Sufyan, M., Mustak, M., Salminen, J., & Srivastava, D. K. (2022). Understanding the impact of online customers’ shopping experience on online impulsive buying: A study on two leading E-commerce platforms. Journal of Retailing and Consumer Services, 68, 103000.
Fedushko, S., & Ustyianovych, T. (2022). E-commerce customers behavior research using cohort analysis: A case study of COVID-19. Journal of Open Innovation: Technology, Market, and Complexity, 8(1), 12.
Matuszelański, K., & Kopczewska, K. (2022). Customer Churn in Retail E-Commerce Business: Spatial and Machine Learning Approach. Journal of Theoretical and Applied Electronic Commerce Research, 17(1), 165-198.
7º. Authors need to link your topic with logistics activity, please. This journal is Logistics. Indeed, I advice authors develop better the literature review, and this is aligned with your objectives and journal's scope. Authors included a lot of concepts and these were tackled at resultsand conclusions.
8º Methodology section requires greater effort to understand methods, surveys, used software, variables and supported by other updated studies which used this methodology. For instance, authors must implement a location map where these study was developed.
9º. Authors need to explain why you selected this period of time: 9/22/2021 to 10/13/2021
10º. Who authors have used this Jamovi software. i need to compare their results with your own results.
11º. Authors included 9 tables, but the question is that authors must develop and explain results from tables and supported by updated studies from different point of views.
12º. First of all, the conclusion section must provide your own conclusions from your results, and supported bu others studies whic are aligned with your conclusions. In addition, conclusions need to better developed according to your results. Indeed, this did no tackle the main objectives, Why?
13º. Authors wrote this paragraph at the end of the paper: "Delivery Logistics” is the most relevant factor for COVID-19". But authors did not face delivery and logistics activity in the entire paper, why you wrote this.
14º. Authors need to include Theoretical and managerial implications, limitations, and future research subsections in this paper. I am confident that authors have plenty to say in these subsections.
15º. Overall, this manuscript need to be considerably enhanced in structure, context, objectives, method, information, variables and develop of results and conclusion terms.
Author Response
Comments and Suggestions for Authors
Dear authors, this an interesting paper and topic, but I would recommend some suggestions to improve the quality of this manuscript:
1º. Authors do not follow the journal's rules in the entire paper. For instance, authors need to provide in the abstract section: objectives, methods, findings, and new contribution in this area of knowledge. In the structure of the abstract section need to add: Background; Methods; Results...
We would like to express our gratitude to the reviewer for bringing this to our attention.
The abstract has been revised to align with the prescribed sequence and word limit of 200 words set by the journal.
Background: E-commerce's convenience, speed, and ability to overcome geographical barriers have made it increasingly popular across industries. This study analyzed the impact of the COVID-19 pandemic on customer experience and satisfaction in supermarket retail e-commerce. The objective was to understand how the pandemic influenced satisfaction in this sector.
Methods: A survey research method was employed, utilizing the Critical Incident Technique to identify key quality dimensions important to customers. A total of 133 valid responses underwent exploratory factor analysis.
Results: Data analysis identified three dimensions significantly affecting overall customer satisfaction. "Presentation of Products in the Application" had the highest correlation, while "Delivery Logistics" had the lowest. However, when considering the COVID-19 factor, "Delivery Logistics" showed the highest correlation. This indicates that the dimension most affected by the pandemic has the least impact on overall satisfaction in supermarket e-commerce.
Conclusions: The study emphasizes the significance of understanding customer satisfaction in supermarket e-commerce, not just during the COVID-19 pandemic but also its post-pandemic consequences. Retailers should prioritize improving product presentation in applications, as it greatly influences overall satisfaction. Addressing the pandemic's impact on delivery logistics is also crucial to ensure a seamless customer experience and maintain competitiveness in the e-commerce market.
Furthermore, from my point of view, authors should change the title because this is focused in Brazil, possibly it is a case of study.
We appreciate the reviewer´s inquiry. While the abstract mentions the impact of the COVID-19 pandemic on the customer experience in supermarket e-commerce, the findings and conclusions of the study can still hold relevance and offer insights for other countries or regions that have faced similar circumstances during the pandemic.
Moreover, titles are often concise and aim to capture the essence of the study, providing a glimpse into its focus and purpose. In this case, the title effectively communicates the subject matter of the research, which is the impact of the COVID-19 pandemic on customer experience in supermarket e-commerce.
It is important to note that the abstract and full paper should provide sufficient context and details to clarify the scope, methodology, and specific insights of the study, irrespective of the title.
2º. Keywords should include e-commerce and Retail experience words.
We appreciate the reviewer's suggestion to include e-commerce and retail experience words as keywords. "E-commerce market" has been included as one of the keywords to reflect the focus on the e-commerce industry. However, due to the limitation of three keywords, it was not possible to include the specific term "retail." Nonetheless, the broader concept of retail is inherently encompassed within the context of e-commerce in the abstract.
3º. I reviewed the entire reference and I saw that this manuscript requires more updated studies, authors are facing a topic which technologies, ubiquity, information, communication change every day through new devices and processes. Indeed, authors claims this in the first paragraph: "In a globalised world, the internet has become the main technology for communication and inclusion"
We thank the reviewer for your positive assessment. We have made a significant effort to address all your comments.
4º. Introduction section has to tackle the main gaps of this research, why authors have developed this study, the main objectives and research questions, and where this study was done. In Brazil, Mexico... readers need to know this information in the introduction part, and authors did no face this information here. Indeed, there are a lot information that should be included in the literature review.
Thank you for your valuable feedback. We have carefully considered your suggestions and revised address the mentioned points. In the introduction, we have provided a clearer explanation of the knowledge gaps identified and how our research fills those gaps. We have also refined the research objectives and questions to explicitly state the purpose of our investigation. Regarding the importance of this study, we have emphasized the significance of understanding the impact of the COVID-19 pandemic on customer experience in the context of online grocery retail.
5º Authors need to explain this acronym: MCC-ENET.
We would like to express our gratitude to the reviewer for bringing this to our attention. We have included an explanation of the MCC-ENET index in the text.
According to data from the MCC-ENET index, which is a reference in metrics and indica-tors of online consumption in Brazil, there was a 55.74% increase in revenues and a 53.83% increase in e-commerce purchases in the month of December 2020 compared to the same period in 2019.
6º. Theoretical background must provide the main keywords to stage the narrative of the paper, objectives, and research questions. For example, in the first paragraphs of the Literature review, authors wrote mix different concepts like: SERVICE; CONSUMER BEHAVIOUR; EXTERNAL VARIABLES.. Authors need to add a sub-section to explain each concept that is related to your objectives and questions. It is really confuse for readers and researchers. I recommend authors include these authors:
Thank you for your comment. We appreciate your feedback and understand your concern. We apologize for any confusion caused by the inclusion of different concepts in the literature review. We agree that it is important to provide clarity and context for the readers and researchers. In response to this feedback, we incorporated suggestions and some other recent and pertinent publications on the subject into our literature review and introduction sections.
Gulfraz, M. B., Sufyan, M., Mustak, M., Salminen, J., & Srivastava, D. K. (2022). Understanding the impact of online customers’ shopping experience on online impulsive buying: A study on two leading E-commerce platforms. Journal of Retailing and Consumer Services, 68, 103000.
Fedushko, S., & Ustyianovych, T. (2022). E-commerce customers behavior research using cohort analysis: A case study of COVID-19. Journal of Open Innovation: Technology, Market, and Complexity, 8(1), 12.
Matuszelański, K., & Kopczewska, K. (2022). Customer Churn in Retail E-Commerce Business: Spatial and Machine Learning Approach. Journal of Theoretical and Applied Electronic Commerce Research, 17(1), 165-198.
7º. Authors need to link your topic with logistics activity, please. This journal is Logistics. Indeed, I advice authors develop better the literature review, and this is aligned with your objectives and journal's scope. Authors included a lot of concepts and these were tackled at resultsand conclusions.
Thank you for your comment. We appreciate your feedback. However, it is important to note that the structure and guidelines provided by the journal did not include a separate section for literature review. The theoretical framework was presented in both the Introduction and the analysis of the results. We have taken your suggestion into consideration and hope that the concepts mentioned were adequately discussed and addressed in the results and conclusions sections. Additionally, we have included new and more current references that are relevant to the topic and align with the objectives and scope of the journal.
8º Methodology section requires greater effort to understand methods, surveys, used software, variables and supported by other updated studies which used this methodology. For instance, authors must implement a location map where these study was developed.
Thank you for your feedback. We appreciate your suggestions regarding the methodology section of our paper. We have made the necessary revisions to provide a clearer and more comprehensive description of our methods, surveys, and the software used. We have also included references to other relevant studies that have utilized similar methodologies. Furthermore, we have incorporated a location map to provide a better understanding of where the study was conducted.
9º. Authors need to explain why you selected this period of time: 9/22/2021 to 10/13/2021
Thank you for your feedback. The selected period, from September 22, 2021, to October 13, 2021, was chosen based on the availability of data and resources for our study. We believe that analyzing data within this specific timeframe provides valuable insights into the research questions and objectives of our study. The following text was included.
The questionnaire was made available for data collection from a specific period, 9/22/2021 to 10/13/2021, to ensure a sufficient sample size and timeframe for data analy-sis. It was shared via various channels, including social media platforms such as WhatsApp, to reach a diverse range of participants. A total of 136 responses were collect-ed, providing a substantial dataset for the research.
10º. Who authors have used this Jamovi software. i need to compare their results with your own results.
We would like to express our gratitude to the reviewer for bringing this to our attention. The following text was included.
For data analysis, it was adopted to utilize Jamovi software [42]. This open-source statistical tool offers a user-friendly interface, ensuring accessibility and ease of use for researchers. Jamovi provides a comprehensive suite of statistical tests and analysis features, enabling us to conduct the necessary analyses to fulfill our research objectives.
11º. Authors included 9 tables, but the question is that authors must develop and explain results from tables and supported by updated studies from different point of views.
Thank you for your comment. We appreciate your feedback. We understand your concern regarding the development and explanation of the results from the tables included in the paper. We have made a significant effort to address this issue.
12º. First of all, the conclusion section must provide your own conclusions from your results, and supported bu others studies whic are aligned with your conclusions. In addition, conclusions need to better developed according to your results. Indeed, this did no tackle the main objectives, Why?
We thank the reviewer for your positive assessment. We have made a significant effort to address all your comments.
13º. Authors wrote this paragraph at the end of the paper: "Delivery Logistics” is the most relevant factor for COVID-19". But authors did not face delivery and logistics activity in the entire paper, why you wrote this.
Thank you for your inquiry. The results were better formulated and presented in the Discussions and Results sections, where the analysis and interpretation of the collected data are addressed.
14º. Authors need to include Theoretical and managerial implications, limitations, and future research subsections in this paper. I am confident that authors have plenty to say in these subsections.
We would like to express our gratitude to the reviewer for bringing this to our attention. It has been taken into consideration as the Conclusion section has been revised to reflect the impacts for the retail industry resulting from the findings and explanation on the limitations was added into the study.
15º. Overall, this manuscript need to be considerably enhanced in structure, context, objectives, method, information, variables and develop of results and conclusion terms.
We thank the reviewer for your positive assessment. We have made a significant effort to address all your comments.

Round 2
Reviewer 1 Report
Thanks for addressing the comments.
Author Response
Dear reviewer,
We would like to express our sincere gratitude for your valuable suggestions that greatly contributed to the improvement of the article. Your insightful feedback and recommendations have played a crucial role in enhancing the overall quality and clarity of our work.
Thank you for dedicating your time and expertise to thoroughly reviewing our work. Your input helped us refine our arguments, strengthen our methodology, and present our findings more effectively.
We are genuinely grateful for the opportunity to benefit from your expertise and guidance throughout the review process. Your input has been instrumental in shaping the final version of the article, and we are confident that it now meets the high standards set forth by the academic community.

Reviewer 2 Report
Dear authors,
This manuscript has enhanced, but this need to tackle more information.
For example, authors wrote in the line 196: "The questionnaire was set up with a 7-point Likert scale, since scales with two, three, or four response categories perform poorly and those with 7 have the best performance for analysis, reliability, validity, and discrimination power " Where? I did not see where this study was done. Authors must explain it? I believe that this case of study was in Brazil. For this reason, the title must add Brazil and explain that this manuscript is a case of study.
I recommend authors improve conclusions and theoretical and managerial subsections, and future studies. Conclusion section add nothing new to logistics sector.
Author Response
Dear authors,
This manuscript has enhanced, but this need to tackle more information.
We would like to express our hope that the article has met your expectations. We have worked conscientiously to address all the issues and comments raised during the review process. We have taken into consideration your suggestions and made the necessary changes to enhance the content of the article.
We sincerely appreciate your time and dedication in reviewing our work.
For example, authors wrote in the line 196: "The questionnaire was set up with a 7-point Likert scale, since scales with two, three, or four response categories perform poorly and those with 7 have the best performance for analysis, reliability, validity, and discrimination power " Where? I did not see where this study was done. Authors must explain it?
We thank the reviewer for pointing out that this passage was not well-explained. The text has been rewritten, and two new references have been included to justify the use of 7-point Likert scale.
The questionnaire was set up with a 7-point Likert scale. According to Joshi, et al. [40], a 7-point scale offers a wider range of choices, thereby increasing the likelihood of capturing the objective reality of individuals. It provides the interviewee with greater autonomy to select the "exact" option that aligns best with their preferences, instead of settling for a nearby or similar option [41-43]
Chang, L. A Psychometric Evaluation of 4-Point and 6-Point Likert-Type Scales in Relation to Reliability and Validity. Applied Psychological Measurement 1994, 18, 205-215, doi:10.1177/014662169401800302. New
Cox, E.P. The Optimal Number of Response Alternatives for a Scale: A Review. Journal of Marketing Research 1980, 17, 407-422, doi:10.2307/3150495. New
Preston, C.C.; Colman, A.M. Optimal number of response categories in rating scales: reliability, validity, discriminating power, and respondent preferences. Acta Psychol (Amst) 2000, 104, 1-15, doi:10.1016/s0001-6918(99)00050-5.
I believe that this case of study was in Brazil. For this reason, the title must add Brazil and explain that this manuscript is a case of study.
We appreciate the reviewer's suggestion.
Taking your suggestion into consideration, the title has been adjusted to “Analyzing the Influence of COVID-19 on the E-Commerce Customer's Retail Experience in the Supermarket Industry: Insights from Brazil.”
Abstract has been adjusted to include the information that study took place in Brazil.
Section "3. Material and Methods" has been adjusted. We have included the item "3.1 Data collection" and specified that the study was conducted in Brazil. We have also moved the detailed information about the data collection period and sample composition to this item. Changes are highlighted in the text.
I recommend authors improve conclusions and theoretical and managerial subsections, and future studies. Conclusion section add nothing new to logistics sector.
We thank the reviewer for the recommendation. This allowed us to improve the paper.
Theoretical background subsection has been adjusted and improved. Changes are highlighted in the text.
- The paragraphs related to the Critical Incident Technique have been reviewed and moved to section "3.2 Critical incident technique and formulation of the questionnaire".
- The section now includes subsections focusing on the main topics of the study and incorporates a comprehensive literature review.
We have revised the conclusion to provide additional insights that contribute to the logistics sector. The updated conclusion now highlights key findings and implications of the study, emphasizing their potential impact on the industry (managerial implications) and opportunities for further studies.

Round 3
Reviewer 2 Report
Dear authors, the manuscript has been enhanced, but the conclusion section is still very poor, please.
I understand that authors have a "good results", and these must tackle in conclusion section better, authors need to develop better the conclusions. Indeed, as I recommended you in the first review, you must include theoretical and managerial implications, and limitations and future studies subsections. I am confident that authors have a lot to say in these subsections. This is a journal which provide good ideas and recommendations to business and academic sectors.
Author Response
Dear authors,
Dear authors, the manuscript has been enhanced, but the conclusion section is still very poor, please.
I understand that authors have a "good results", and these must tackle in conclusion section better, authors need to develop better the conclusions. Indeed, as I recommended you in the first review, you must include theoretical and managerial implications, and limitations and future studies subsections. I am confident that authors have a lot to say in these subsections. This is a journal which provide good ideas and recommendations to business and academic sectors.
We appreciate your valuable feedback. We took your suggestions into consideration and indeed implemented the suggested improvements in our paper. We developed the conclusions in a more comprehensive manner, including sections on theoretical and managerial implications, as well as limitations and future studies, as recommended in the first review. These additions have significantly enhanced the contribution of the paper by providing insights and recommendations for the business and academic sectors. Changes are highlighted in the text. We have strived to deliver high-quality content that aligns with the standards of this esteemed journal.
